# Skin graft with dermis and appendages generated in vivo by cell competition

Hisato Nagano [1,2,3,9], Naoaki Mizuno [1,2,4,9] ✉, Hideyuki Sato[1,2], Eiji Mizutani [1,2,5], Ayaka Yanagida [1,2,6], Mayuko Kano [1,2,7], Mariko Kasai[1,2], Hiromi Yamamoto[1,2], Motoo Watanabe[1,2], Fabian Suchy [8], Hideki Masaki[1,2] & Hiromitsu Nakauchi [1,2,8] ✉

Autologous skin grafting is a standard treatment for skin defects such as burns. No artificial skin substitutes are functionally equivalent to autologous skin grafts. The cultured epidermis lacks the dermis and does not engraft deep wounds. Although reconstituted skin, which consists of cultured epidermal cells on a synthetic dermal substitute, can engraft deep wounds, it requires the wound bed to be well-vascularized and lacks skin appendages. In this study, we successfully generate complete skin grafts with pluripotent stem cell-derived epidermis with appendages on p63 knockout embryos' dermis. Donor pluripotent stem cell-derived keratinocytes encroach the embryos' dermis by eliminating p63 knockout keratinocytes based on cell-extracellular matrix adhesion mediated cell competition. Although the chimeric skin contains allogenic dermis, it is engraftable as long as autologous grafts. Furthermore, we could generate semi-humanized skin segments by human keratinocytes injection into the amnionic cavity of p63 knockout mice embryos. Niche encroachment opens the possibility of human skin graft production in livestock animals.

Skin is mainly composed of the epidermis and dermis, which are connected by abasement membrane. In early developmental stages, the epidermis is a single layer of ectodermal progenitor cells named the surface ectoderm. The surface ectoderm develops into the epidermis through differentiation and stratification during development. Skin appendages, including hair follicles, sweat glands, and nails, are derived from the surface ectoderm. Generating functional and transplantable skin with skin appendages is one of the ultimate goals of regenerative medicine to solve the skin graft shortage[1]. The development of keratinocyte culture and in vitro tissue engineering techniques have partially reproduced the skin. Of these skin substitutes, cultured epidermis[2] has been the most successful clinically. However, cultured epidermis does not have a dermal layer and will seldom be engrafted on deep wounds caused by severe burns[3]. Alternatively, reconstituted skins are tissue-engineered skin substitutes composed of cultured keratinocytes and a synthetic pseudo-dermis. They are more similar to autologous skin grafts, but reconstituted skin will rarely engraft the wound bed without excellent

[1]Division of Stem Cell Therapy, Center for Stem Cell Biology and Regenerative Medicine, Institute of Medical Science, University of Tokyo, 4-6-1 Shirokanedai, Minato-ku, Tokyo 108-8639, Japan. [2]Stem Cell Therapy Laboratory, Advanced Research Institute, Tokyo Medical and Dental University, 1-5-45 Yushima, Bunkyo-ku, Tokyo 113-8510, Japan. [3]Department of Plastic and Reconstructive Surgery, National Defense Medical College, 3-2 Namiki, Tokorozawa, Saitama 359-8513, Japan. [4]Department of Experimental Animal Model for Human Disease, Center for Experimental Animals, Tokyo Medical and Dental University, 1-5-45 Yushima, Bunkyo-ku, Tokyo 113-8510, Japan. [5]Laboratory of Stem Cell Therapy, Institute of Medicine, University of Tsukuba, 1-1-1 Tennodai, Tsukuba, Ibaraki 305-8577, Japan. [6]Department of Veterinary Anatomy, The University of Tokyo, Yayoi 1-1-1, Bunkyo-ku, Tokyo 113-8657, Japan. [7]Metabolism and Endocrinology, Department of Medicine, St. Marianna University School of Medicine, 2-16-1 Sugao, Miyamae-ku, Kawasaki, Kanagawa 216-8511, Japan. [8]Institute for Stem Cell Biology and Regenerative Medicine, Stanford University School of Medicine, Stanford, CA 94305, USA. [9]These authors contributed equally: Hisato Nagano, Naoaki Mizuno. ✉e-mail: nmizuno1.sct@tmd.ac.jp; nakauchi@stanford.edu

vascularization[4]. Furthermore, they do not fully recapitulate native skin because they lack skin appendages[5]. Recently, human skin organoids have been reported[6], and hair follicles and sebaceous glands have been generated. However, these organoids are cystic and small (500–1000 μm in size) and have not yet been developed into skin substitutes that could immediately restore the skin barrier function. As yet, no skin substitutes are equivalent to autologous skin grafts[7]. To overcome these problems, we generated skin grafts composed of the fully stratified epidermis with appendages on thick dermis in vivo, which might result in comparable engraftment and function of autologous skin grafts.

## Results

### PSC-derived epidermis and appendages generated on chimeric dermis by p63 knockout of host embryos

For the proof-of-concept of in vivo skin generation, we aimed to generate pluripotent stem cell (PSC)-derived skin through the systemic chimera formation of donor mouse PSCs and p63 knockout mouse embryos. p63 is essential for the differentiation of stratified epithelium, including keratinocytes[8,9]. Surface ectoderm, epidermal progenitor cells at mid-organogenesis stage, fail to stratify normally in p63 knockout mice. Though p63 knockout surface ectoderm can remain on the fetal body surface at birth[10], it cannot contribute to mature cells of the epidermis nor appendages in a systemic chimera of wild-type PSC and p63 knockout host embryos.

We prepared p63 knockout mouse embryos by clustered regularly interspaced short palindromic repeats/CRISPR-associated protein 9 (CRISPR-Cas9) genome editing. There are two major isoforms of p63: TA63 and ΔNp63. We designed gRNA on exon 5, which codes the DNA binding domain common to the two isoforms (Fig. 1a). Cas9 ribonucleoprotein (RNP) was introduced into zygotes by electroporation, and the embryos were transferred into the pseudo-pregnant mice to obtain p63 knockout fetuses (Supplementary Fig. 1a–c). As previously reported[8,9], the p63 knockout fetuses showed translucent skin, shortened limbs due to apical ectodermal ridge deficiency[11], and maxillofacial deformities (Fig. 1b and Supplementary Fig. 1d). Seventeen of 22 live fetuses showed the knockout phenotype, and 11 of 17 phenotypically Knockout fetuses had bi-allelic loss-of-function genotypes (Supplementary Fig. 1a). Although multiple gRNAs could achieve higher editing efficiency[12], we introduced single gRNAs because small InDels are precisely quantified by simple genotyping methods (Supplementary Fig. 2a–d). Some fetus showed knockout phenotypes even though their genotype contained small in-frame mutations, probably because the gRNA was designed to target a core domain of p63 (Supplementary Fig. 1b)[13]. However, we excluded these fetuses from the following experiments because their genotype is not guaranteed to produce a knockout phenotype.

Next, we generated the systemic chimera comprised of allogenic mouse embryonic stem cells (ESCs) and p63 knockout embryos. The mouse ESCs, which were labeled with an enhanced green fluorescent protein (EGFP), were microinjected into the p63 knockout preimplantation embryos at the 4–8 cell stage. The chimeric blastocysts were transferred into the uterus of pseudo-pregnant mice at the blastocyst stage (Fig. 1c). Hereafter, chimeras generated with p63 knockout embryos and donor PSCs are referred to as p63 knockout chimeras.

We obtained eight p63 knockout chimeric neonates from 79 transferred embryos (10.1%) (Supplementary Fig. 1e). We quantified donor cell contribution to the skin and the other tissues in p63 knockout chimeras. The skin of p63 knockout chimera was harvested and treated with dispase. The isolated epidermis was dissociated with trypsin/EDTA and analyzed by flow cytometry. We found that almost 100% of the keratinocytes, defined by CD45−/CD117−/CD49f+ cells[14], were positive for EGFP (Fig. 1d). By contrast, the donor ESCs contributed to splenocytes and melanocytes (neural crest cells derived CD45−/CD117+ cells in the skin) at the statistically lower frequency in p63 knockout chimera than those to the skin keratinocytes (Fig. 1e).

The total skin area of p63 knockout chimeras at E18.5 increased logarithmically with global chimerism assumed by the splenocyte chimerism[15]. It was estimated that 1.6% global chimerism was sufficient for 50% body surface coverage (Fig. 1f and Supplementary Fig. 1f). Some fetuses showed normal limbs and craniofacial formation. In the restored forearms, the epidermis was comprised of donor PSC-derived cells, while subcutaneous and bone tissues were chimeric (Supplementary Fig. 3a–i).

Histopathological analysis revealed that the donor PSC-derived keratinocytes of p63 knockout chimeras formed a well-differentiated and stratified epidermis similar to the native keratinocytes of wild-type (WT) mice (Fig. 1g). The hair follicles were also derived from donor PSCs (Fig. 1h, i).

The surface ectoderm remnant (SER), which is the undifferentiated epidermis of p63 knockout mice, remains at birth. However, it is fragile and easily detached[10]. Since the epithelial differentiation of the p63 knockout embryos does not proceed normally, the SER of p63 knockout fetuses does not express KRT14, which is expressed in basal cells of WT mice (Supplementary Fig. 4a, b).

As CD49f is also a marker of mature keratinocytes, the host embryo-derived SER in the skin formed by p63 knockout chimeras is undetectable by flow cytometry. We found that SER is continuously positive throughout embryogenesis for KRT8/18, which is a marker for the normal surface ectoderm at E8.5 (Supplementary Fig. 4b). Histological analysis of p63 knockout chimeras at E18.5 reveals that there are no KRT8/18+EGFP− cells within the epidermis, but there are outside the stratum corneum (Supplementary Fig. 4c). These results showed that the stratified epidermis of p63 knockout chimeras was totally derived from donor PSCs and that the host-derived SER is absent inside the epidermis.

### Cell competition promotes large continuous sheets of PSC-derived keratinocytes devoid of p63 knockout keratinocytes

p63 knockout chimeras have a blend of normal skin regions (from PSC-derived keratinocytes) and abnormal skin regions containing the host SER. If the PSC-derived skin patches are frequently interrupted by abnormal host skin, we could only harvest small skin fragments instead of a transplantable skin sheet. Indeed, the host embryos' skin finely intermingles with PSC-derived skin in wild-type chimeras at E18.5 (Fig. 2a).

According to our initial hypothesis, the donor PSC-derived skin regions in p63 knockout chimeras at E18.5 would have cleft-like ulcers after the mechanical detachment of SERs. We expected such clefts would be covered by surrounding PSC-derived keratinocytes through wound healing after birth (Supplementary Fig. 5a, b). However, the donor PSC-derived skin regions in p63 knockout chimeras were not finely disperse, but rather large segmented clusters even before birth (Fig. 2b). Therefore, we proposed a new hypothesis: the host embryo-derived SERs were lost from an early developmental stage, and donor PSC-derived epidermal cells became dominant in the p63 knockout chimera.

Keratinocytes derived from host embryos and donor PSCs were randomly distributed and finely intermingled in wild-type chimeras at E14.5 as well as at E18.5 (Fig. 2c and Supplementary Fig. 5b). On the other hand, in p63 knockout chimeras at E14.5 donor PSC-derived cells were located on the basal side of host embryo-derived keratinocytes at the boundary (Fig. 2d and Supplementary Fig. 5b). We statistically confirmed that PSC-derived keratinocytes form more contiguous segments than the random distribution expected in each p63 knockout chimera (Supplementary Fig. 5c–g). These data indicated that PSC-derived keratinocytes competitively takeover the epidermal niche on dermis by undermining p63 knockout SERs, resulting in segmented clusters without interspersed SER partitions. We named this

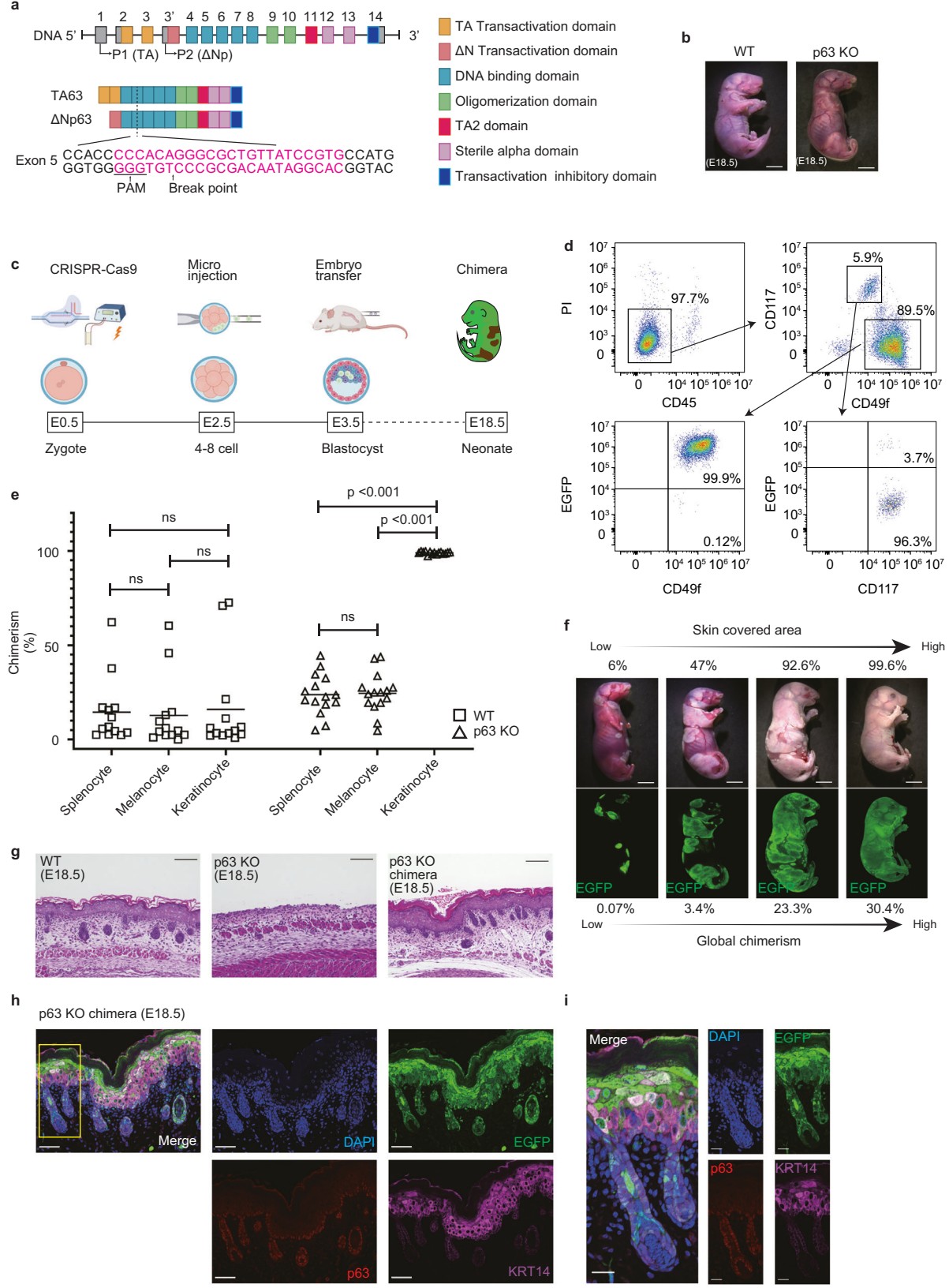

phenomenon "niche encroachment". Host SER survived and remained until birth in the p63 knockout embryo, while SER was selectively eliminated by coexisting donor PSC-derived cells in p63 knockout chimeras. This meets the definition of cell competition[16], during which the loser cells could be eliminated by mechanical means or by direct induction of apoptosis. Therefore, we performed TUNEL staining to

verify whether SER's cell-competitive detachment is apoptosis-dependent. The number of TUNEL-positive cells in p63 knockout embryos and p63 knockout chimeras was not significantly different from the number of TUNEL-positive cells in WT embryos, suggesting that SER was detached in an apoptosis-independent manner (Supplementary Fig. 6a–l). Since we confirmed that apoptosis-independent

**Fig. 1 | PSCs-derived epidermis and appendages generated on chimeric dermis by p63 knockout of host embryos. a** CRISPR-Cas9 gRNA design for p63 gene. **b** Macroscopic images of a WT (left) mouse and a p63 knockout mouse (right). Scale bars, 5 mm. **c** Experimental schematic for p63 knockout chimera generation (E9.5 to E18.5 by C-section). Created with BioRender.com. **d** Flow cytometry of p63 knockout chimera epidermis (E18.5). **e** The comparison of chimerism between WT chimera and p63 knockout chimera (Splenocyte; CD45⁺, Melanocyte; CD45⁻CD117⁺, Keratinocyte; CD45⁻CD117⁻CD49f⁺). Statistical analysis was performed using two-sided Friedman tests followed by Dunn's test with correction for multiple comparisons. Square, WT chimera (*n* = 13); triangle, p63 knockout chimera (*n* = 15).

**f** Macroscopic and fluorescence (EGFP) images of four p63 knockout chimera (E18.5) with different global chimerism (splenocytes). The upper number is skin-covered areas, and the lower number is global chimerism in each chimera. Scale bars, 5 mm. **g** Histological staining of WT, p63 knockout, and p63 knockout chimera skin. Scale bar, 100 μm. **h** Immunofluorescence staining of the skin of p63 knockout chimera. DAPI, nuclear; EGFP, the label of donor cell; p63, basal cell marker; KRT14, mature keratinocytes in the basal layer. Scale bars, 50 μm. **i** Magnified hair follicle image of yellow box in (**h**). Scale bars, 20 μm. Source data are provided with this paper.

cell competition in the epidermis of p63 knockout chimeras eliminates p63 knockout keratinocytes by niche encroachment, we sought to elucidate further molecular mechanisms. The SER in p63 knockout embryos is known to be easily detached at birth[10], possibly due to an adhesive failure at the epidermal-dermal junction[17]. Because SERs appear less adhesive to the basement membrane than WT keratinocytes during skin organogenesis, we hypothesized that host SERs may be pushed out from the niche by horizontally proliferating PSC-derived keratinocytes in p63 knock chimeras.

To verify this hypothesis, we performed single-cell RNA-Seq of SER and WT keratinocytes at E14.5 (Supplementary Data 1). Although WT keratinocytes were collected with high purity, SERs were highly contaminated with dermal cells. Because KRT8 expression of SERs at E14.5 was confirmed by immunohistochemistry, we performed qPCR-based enrichment of *Krt8*-positive samples and selectively sequenced them (Supplementary Fig. 7a). Cell lineages and subtypes were manually annotated based on the marker gene expression (Supplementary Fig. 7b, c). PCA analysis showed that WT keratinocytes and SER formed different clusters for each genotype and cell subtype (Supplementary Fig. 7d). Gene expression of hemidesmosome and focal adhesion were then compared for the basal keratinocytes of each genotype because they are two major structures responsible for the epidermal adhesion to the dermis. Hemidesmosome is mainly composed of keratin 5 (KRT5), keratin 14 (KRT14), dystonin (DST), plectin (PLEC), collagen type XVII alpha 1 chain (COL17A1), integrin alpha 6 (ITGA6), integrin beta 4 (ITGB4), and laminin 332. Most of those hemidesmosome-related genes were down-regulated in the basal keratinocytes of the p63 knockout embryos (p63 KO-SER) compared to the basal keratinocytes of wild-type embryos (WT-KCB) (Fig. 2e, f and Supplementary Table 1). As for focal adhesion-related genes, only *Fermt1* and *Itga3* were down-regulated in the p63 knockout-SER group compared to the WT-KCB group (Supplementary Fig. 7e, f and Supplementary Table 1). Collectively, these data suggest that the primary mechanism of SER cell displacement arises from the incomplete hemidesmosome of p63 knockout keratinocytes.

### Skin grafts generated in p63 knockout chimeras demonstrate long-term engraftment despite the allogeneic dermis

Next, we investigated whether the skin generated from p63 knockout chimera could be engrafted for a long time after orthotopic skin transplantation. Skin grafts from PSC-derived skin segments also contain host-derived cells, including melanocytes, fibroblasts, and endothelial cells. Although the keratinocytes are autologous, MHC-mismatched components in chimeric dermis could be rejected, potentially resulting in total graft failure after skin grafting. To confirm whether such semi-autologous skin grafts could be engraftable, we compared graft survival on recipient mice of different genetic backgrounds.

Since EGFP is highly antigenic and is considered to be a factor of immune rejection during skin grafting[18], we generated new host p63 knockout embryos by crossing C57BL/6-Tg (CAG-EGFP) and DBA/2. These F1 hybrid embryos were then injected with C57BL/6N-derived mouse ESCs (Fig. 3a). Skin grafts collected from these p63 knockout chimeras would be semi-autologous when transplanted onto

C57BL/6-Tg (CAG-EGFP), whereas they are practically autologous when transplanted onto F1 hybrid mice of C57BL/6-Tg (CAG-EGFP) and DBA/2. Surprisingly, the semi-autologous skin grafts were engrafted for more than 3 months after transplantation, comparable to autologous transplantations (Fig. 3b–d). The hair growth within the graft area was similar to the skin in the recipient mice (Fig. 3b). On the other hand, chimeric skin grafts with intermingled allogenic epidermal cells were rejected within 14 days (Fig. 3d, allogenic). To exclude the possibility of a high graft survival rate achieved by high dermal cell chimerism, we selected fetuses with dermal chimerism in the 20–40% range in WT and p63 knockout chimera. The subgroup analysis showed that the graft survival period was predominantly longer in the p63 knockout chimera than WT chimera (Supplementary Fig. 8a–c). These results indicate that epidermal MHC matching is sufficient for long-term skin graft engraftment, while MHC matching of the dermal cells is not necessary.

### Semi-humanized skin generation by keratinocytes injection into the amnionic cavity of p63 knockout mice

Finally, we attempted to generate semi-humanized skin by injecting human keratinocyte into the amnionic cavity of p63 knockout mice. Single-cell suspensions ($1 \times 10^7$ cells/500 μl) of EGFP-labeled HaCaT, which is a commonly used human immortal keratinocyte line, were injected into the amniotic cavity of a p63 knockout mouse embryo at E13.5 by a trans-uterus approach (Fig. 4a). We found that injected EGFP-expressing HaCaT formed a segment/sheet-like structure on p63 knockout mouse embryos at E18.5 (Fig. 4b). Histopathological analysis showed that the SERs of the p63 knockout embryo were replaced by multi-layered human keratinocytes adhered to the host dermis, even though the state of HaCaT was in a single cell suspension at the time of injection (Fig. 4c). Differentiation markers such as KRT1/10, LORICRIN, and IVL were positive in the outer layer including the suprabasal layer (Fig. 4d–i), suggesting that HaCaT was stratified and differentiated on the p63 knockout dermis. These results indicate that niche encroachment by intraamniotic-injected human keratinocytes could generate contiguous human skin segments in p63 knockout host embryos.

### Discussion

In this study, we generate PSC-derived full-thickness skin grafts with appendages on p63 knockout mouse dermis by cell-ECM mediated cell competition between donor cells and p63 knockout keratinocytes. We named this technique "niche encroachment" since donor PSC-derived cells invade the host embryos' epidermal niche by eliminating host keratinocytes during embryogenesis. The donor PSCs recapitulated skin organogenesis in vivo and formed skin grafts equivalent to the native skin, which has not previously been achieved by in vitro differentiation. The in vivo-generated skin grafts are semi-autologous with donor keratinocytes and skin appendages on the chimeric dermis. It has been known that skin is highly immunogenic, and allogenic grafts are rapidly rejected after transplant[19]. However, we show that the semi-autologous skin grafts successfully engrafted for a long time, comparable to autologous transplantation, despite the presence of host cells in the dermis and vessels. This may indicate that the strong immunogenicity is mostly restricted to the keratinocytes, not the ECM-

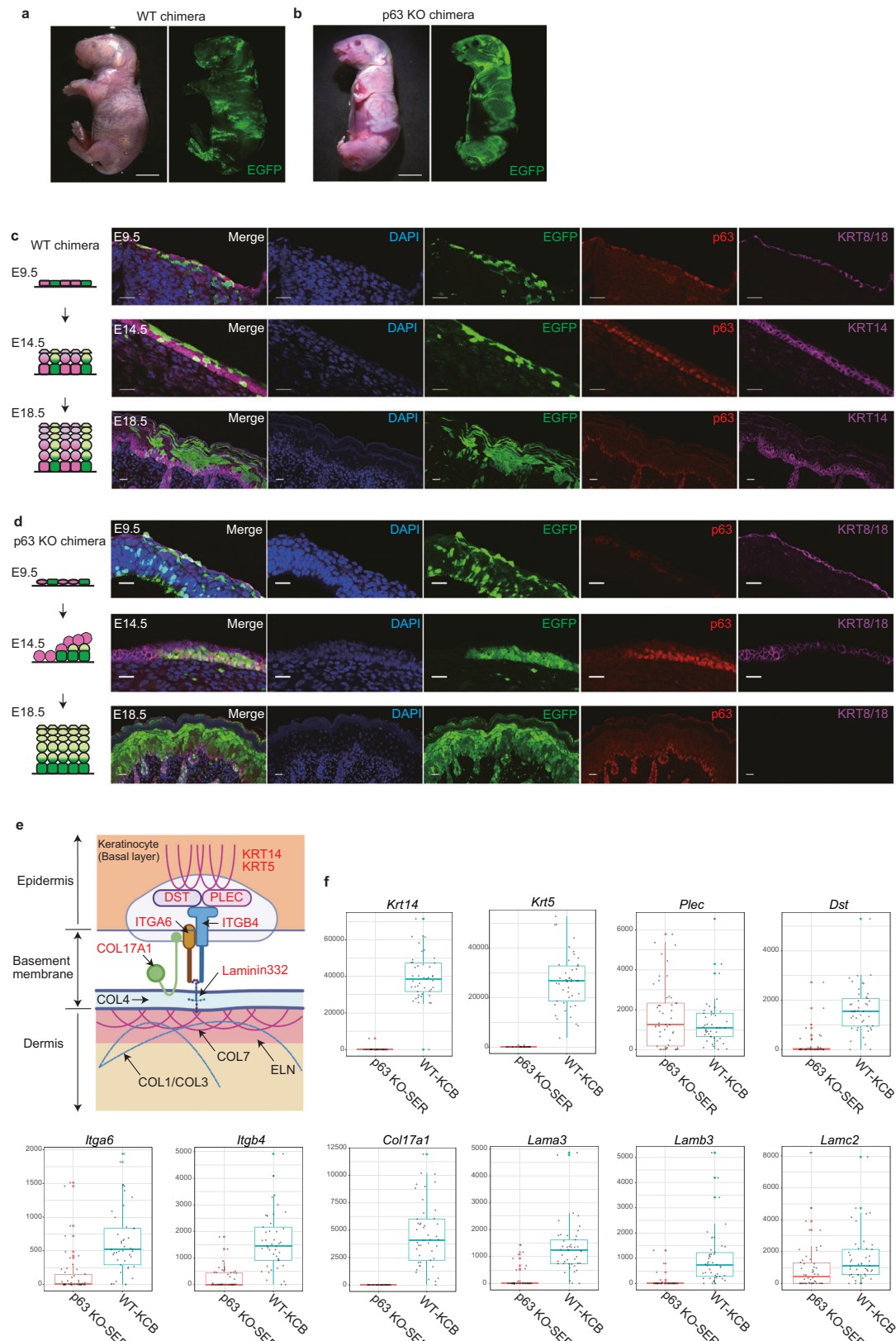

rich dermis. This opens the possibility of human skin graft production in livestock animals.

Autologous skin grafting is the gold standard treatment for skin defects such as burns. Full-thickness skin grafting, on which appendages are conserved in the dermis, is ideal for reconstructing the appearance and functions of the native skin. However, it is necessary to

transfer the autologous skin from a donor site to a recipient site. With extensive body surface area defects, it is difficult to harvest enough skin grafts from the limited donor sites. In such cases, split-thickness grafting is an alternative. Split-thickness grafts are thinner autologous skin grafts containing only partial dermis. This enables the donor sites to heal by keratinocyte proliferation from appendages, and the

**Fig. 2 | PSC-derived keratinocytes form large segmented skin clusters without p63 knockout keratinocytes by cell competition. a, b** Macroscopic and fluorescence (EGFP) images of WT chimera (**a**) and p63 knockout chimera (**b**). Scale bars, 5 mm. **c, d** Skin development/cytokeratin expression pattern (left panels) and immunofluorescence staining (right panels) in WT chimera (**c**) and p63 knockout chimera (**d**). Scale bars 20 μm. DAPI, nuclear; EGFP, the label of donor PSC; p63, basal cell marker; KRT8/18, immature keratinocyte; KRT14, mature keratinocyte. **e** Schematic of the hemidesmosome. Proteins colored red are mainly produced by keratinocytes. Created with BioRender.com. **f** Comparison of gene expression between p63 knockout-SER (Surface ectoderm remnant) and WT-KCB (keratinocyte in basal layer). p63 Knockout-SER, $n = 50$ cells from 7 embryos; WT-KCB, $N = 45$ cells from 3 embryos. Embryos examined over 5 independent experiments. The median is the center bold line, and first and third quartiles are the bounds of box. The whiskers extend from the box and represent the data points that fall within 1.5 times the interquartile range (IQR) from the lower and upper quartiles. Colored points are outliers defined as the points under the first quartile − 1.5 times IQR or the points over the third quartile + 1.5 times IQR. Source data are provided in the Supplementary Data 1.

regenerated skin can be repeatedly transplanted several times from the same site. However, split-thickness skin grafts do not contain appendages and are more prone to shrinkage and scar formation[20] than full-thickness skin grafting, leading to poor functional and cosmetic results.

To overcome these clinical limitations, artificial skin substitutes have been developed. Indeed, cultured epidermis and some of the reconstituted skins are already available commercially, but they are still inferior to autologous skin grafts. Cultured epidermis can engraft only in shallow wounds where the dermis remains as a scaffold. Although reconstituted skins could be applied to deeper ulcers with dermal injury, the engraftment rate is insufficient. In addition, those substitutes generated in vitro lack appendages. The full-thickness skin grafts obtained from artificial substitutes described here would not be restricted to injury severity nor the remaining skin area for graft donation. As such, it has the potential to become the new standard over current treatment options.

There are several approaches to generate PSC-derived organs in vivo. Blastocyst complementation is the approach that we pioneered, in which systemic chimeras are generated by microinjecting donor PSCs into preimplantation embryos of organ-deficient transgenic animals. The targeted organ niche is occupied by donor cells during embryogenesis, resulting in donor-matched functional organs at birth. A number of PSC-derived functional organs have been generated in allogeneic and xenogeneic hosts[21], including pigs[22]. However, some organs are considered difficult targets for blastocyst complementation because inadequate donor PSC contribution to the targeted organ niche generates an incomplete primordium, resulting in later organ failure and embryonic lethality. In contrast, one of the major advantages of niche encroachment is that the donor PSC-derived keratinocytes take over the skin niche throughout embryogenesis, allowing the skin to be gradually replaced without severe organ failure. Although the body surface of the p63 knockout chimera is not fully covered by mature epidermis, donor PSC-derived keratinocytes formed large, segmented skin clusters suitable for skin grafting. Even when global chimerism is low, the purity of keratinocytes of the skin segment is 100% donor derived.

This mechanism of cell competition in the p63 knockout chimeras is due to cell-to-ECM disruption. This is different from the conventional cell competition[16,23] in which cell-cell interaction is the main mechanism. The canonical cell competition directly induced apoptosis of loser cells adjacent to winner cells, but apoptosis independent-cell competition was also reported. Apical extrusion in the luminal organs removes the outlier cells from the contiguous epithelium. However, it is mediated by disruption of cell-cell adhesion.

Interestingly, niche encroachment uses a mechanism similar to the one proposed by Liu et al. They revealed a novel cell competition mechanism in aged mouse skin where senescent keratinocytes are eliminated due to a decreased cell-ECM adhesion arising from downregulation of COL17A1[24]. Here we demonstrate that in the absence of p63, most hemidesmosome-related genes, including *Col17a1*, are downregulated during embryogenesis thus allowing the opportunity for wild type PSC-derived cells to encroach the niche. Thus, lack of p63 plays a role in cell-ECM mediated cell competition in the chimeras. It is of interest whether p63 is down regulated in senescent cells and, if so,

whether increased expression of the p63 gene results in skin rejuvenation.

Revertant mosaicism[25] is occasionally observed in skin disorders such as epidermolysis bullosa. This is similar to niche encroachment. The epithelial cells regain normal gene function through spontaneous somatic mutation and replace the original mutant epithelial cells with congenital loss-of-function mutations. However, revertant mosaicism expands slowly and forms small patches. The spontaneous revertant clones grow onto vacant dermis after pathogenic epithelial disruption by mechanical stress. In contrast, niche encroachment could widely and quickly replace p63 knockout embryos' epidermis during embryogenesis because wild-type donor cells actively eliminate surrounding mutant keratinocytes. Cell competition facilitates more efficient tissue replacement in niche encroachment than the simple growth advantage with external forces observed in revertant mosaicism.

In this study, we successfully generated partially differentiated semi-humanized skin by niche encroachment. Human keratinocyte cell lines, which were injected into the mouse amniotic cavity, engrafted onto the embryos' skin and expanded on the p63 knockout mouse dermis. The human cells formed a sheet-like and multi-layered structure with cytodifferentiation. Niche encroachment is promising even in the xenogeneic condition and during post-implantation organogenesis. Though postnatal immunological responses[26] may also be a problem in xenogeneic chimeras, modulation of genes involved in immunity (*IL2RG*, *RAG2*, etc.) may support efficient organ generation in vivo. While further studies are required using human primary keratinocytes to determine whether the stemness of keratinocytes is maintained or whether the injected keratinocytes could form skin appendages, this method may provide a practical alternative to organ complementation using systemic xeno-chimera formation. As opposed to blastocyst injection, this method of in utero injection avoids the developmental xenobarrier[27] that has been a problem in generating human-animal chimeras with large evolutionary distances. Indeed, others have performed in utero cell injection and it has been shown that human melanocytes survive and function after birth when transplanted into E8.5 post-implantation mouse embryos[28]. However, there have been no reports regarding the generation of the solid organs grown from cells injected in utero. Furthermore, targeted organ generation has the advantage of avoiding ethical concerns[29] about semi-humanized intelligence and human germ cell production in interspecies chimeras. Gene knockouts related to neuronal (*OTX2*) or germinal (*PRDM14*[30]) cell differentiation may further minimize the risk of human stem cell contribution in the brain or gonad.

Generation of skin appendages is initiated around E13[31], but to inject human primary keratinocytes into amnion cavity before this stage is technically challenging. Instead of primary keratinocytes, PSC-derived keratinocytes or skin organoids[6] could be injected in utero and combined with niche encroachment. They are potentially more abundant in multipotent skin stem cells than primary keratinocytes and could be an unlimited source of autologous stem cells. The small body size and the short length of pregnancy in mice limited the timing of injection of human cells and the time available for skin development. Use of livestock animals with a longer gestation period and a larger body size would help generate large human skin grafts with appendages.

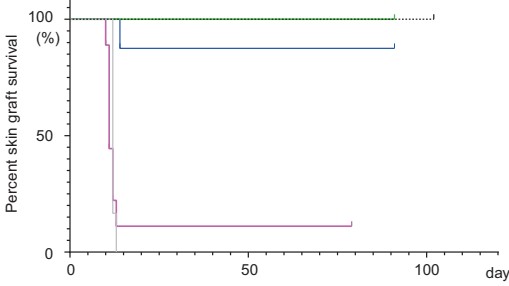

| Group | Symbol color | Sample size | Embryos | Donor cells | Epidermis | Dermis | Recipient |
|---|---|---|---|---|---|---|---|
| Positive control | ···ı·· | 6 | WT (BDF1-EGFP) | (−) | BDF1-EGFP | BDF1-EGFP | BDF1-EGFP |
| Autologous | ⊥ | 10 | p63 KO (BDF1-EGFP) | C57BL/6 | C57BL/6 | Chimeric | BDF1-EGFP |
| Semi-autologous | ⊥ | 8 | p63 KO (BDF1-EGFP) | C57BL/6 | C57BL/6 | Chimeric | C57BL/6-Tg (EGFP) |
| Allogeneic | ⊥ | 9 | WT (BDF1-EGFP) | C57BL/6 | Chimeric | Chimeric | C57BL/6-Tg (EGFP) |
| Negative control | ·ı· | 6 | WT (BDF1-EGFP) | (-) | BDF1-EGFP | BDF1-EGFP | C57BL/6-Tg (EGFP) |

**Fig. 3 | Skin grafts generated in p63 knockout chimera could be engrafted for a long time despite the allogeneic dermis. a** Experimental schematic of semi-autologous skin grafting. Created with BioRender.com. **b** Macroscopic image after semi-autologous transplantation. POD postoperative day. **c** Immunofluorescence staining of the boundary of grafts and recipients in semi-autologous transplantation (upper panel) and autologous transplantation (lower panel) at 91 POD. White arrows indicate the boundary between skin grafts (EGFP⁻) and recipient skin (EGFP⁺). EGFP, host cells; DAPI, nuclear; p63, basal cell marker; KRT14, mature keratinocyte. Scale bars, 20 μm. **d** Experimental groups and graft survival curves of orthotopic skin transplantations. Source data are provided with this paper.

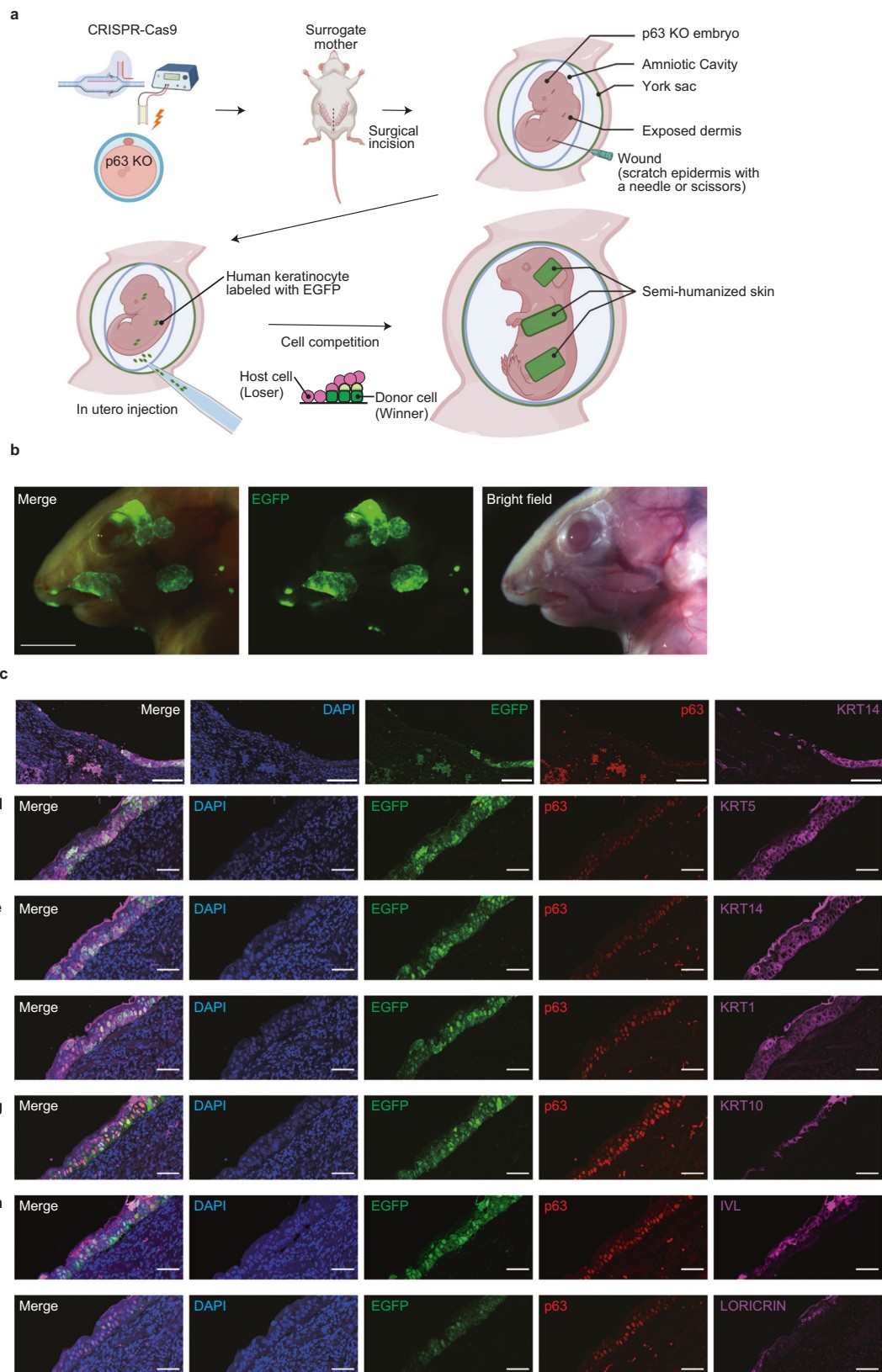

**Fig. 4 | Semi-humanized skin generation by keratinocytes injection into the amnionic cavity of p63 knockout mice. a** Experimental schematic. Created with BioRender.com. **b** Macroscopic and fluorescence images of human skin segment by keratinocytes injection into amnionic cavity. Scale bars, 2.5 mm. **c** Immunofluorescence staining of the boundary between humanized skin segment and SER. DAPI, nuclear; EGFP, the label of HaCaT; p63, stem cell marker; KRT14, basal cell marker. Scale bars, 100 μm. **d–i** Immunofluorescence staining of human skin segment. DAPI, nuclear; EGFP, the label of HaCaT; p63, stem cell marker. Scale bars, 50 μm. **d** KRT14, basal cell marker. **e** KRT10, suprabasal cell marker. **f** IVL, terminal differentiation marker. **g** KRT5, basal cell marker. **h** KRT1, suprabasal cell marker. **i** LORICRIN, terminal differentiation marker.

## Methods

### Ethics declarations

In this study, we injected HaCaT cells to amniotic cavity of E13.5 mouse fetus to transplant human keratinocyte cells, (see "In utero injection" below). This is different from human PSC-derived chimeras with pre-implantation animal embryos, which potentially generate systemic human-animal chimeras. We also used HaCaT cell line as a graft instead of human primary keratinocyte. Therefore, described HaCaT cells-derived chimera study was able to be performed under the approval of the animal care and use committee of the Institute of Medical Science, the University of Tokyo (Permission No. PA21-16, PA21-47)

### Animals

C57BL/6N, ICR, C57BL/6N-Tg (CAG-EGFP), and DBA/2 mice were purchased from SLC Japan (Shizuoka, Japan) and were reared under the prescribed conditions (light: a.m. 8:00–p.m. 20:00/dark: p.m. 20:00–a.m. 8:00). All experiments were performed in accordance with the animal care and use committee guidelines of the Institute of Medical Science, the University of Tokyo (Permission No. PA21-16, PA21-47).

### Cell culture

The ESCs of SGE2[26] (C57BL/6N-Tg(CAG-EGFP) mESC) and B6ES2 (C57BL/6N mESC) were cultured on mitomycin C-(Fujifilm Wako Pure Chemical Corporation, Tokyo, Japan, Catalog No. 139-18711)-treated mouse embryonic fibroblasts in N2B27 medium (Invitrogen, Catalog No. 11330-032, 21103-049, 17502048, 17504044) containing 3.0 μM CHIR99021 (Axon Medchem, Groningen, Netherlands, Catalog No. Axon1386), 1.0 μM PD0325901 (Fujifilm Wako Pure Chemical Corporation, Tokyo, Japan, Catalog No. 162-25291), 1000 U/ml of leukemia inhibitory factor (Millipore, Billerica, MA, Catalog No. ESG1107).

HaCaT (human immortalized keratinocytes, male) were purchased from Cell Line Service (Germany, Product No. 300493) and were cultured in CnT-07 medium (CELLnTEC, Bern, Switzerland, Catalog No. CnT-07) and transduced with a CAG-EGFP expressing lentiviral vector (HaCaT-EGFP).

### Embryo culture, electroporation, and micromanipulation

To obtain ICR zygotes or BDF1-EGFP zygotes (C57BL/6N-Tg(CAG-EGFP) females crossed with DBA/2 males), in vitro fertilization was performed in HTF medium (ARK-Resource, Kumamoto, Japan).

For genome editing, Cas9 RNP electrode with 200 ng/μl p63 sgRNA; CACGGATAACAGCGCCCTGT (Integrated DNA Technologies, Iowa, USA), 100 ng/μl Sp.Cas9 protein (Integrated DNA Technologies, Iowa, USA, Catalog No. 1081059) in Opti MEM (Invitrogen, Carlsbad, CA, Catalog No. 31985-062) was introduced into zygotes by electroporation[32] using the Genome Editor (BEX, Tokyo, Japan). For p63 knockout chimera generation, SGE2 mESCs were injected into the preimplantation ICR embryos at the 4–8 cell stage on F0 generation of the p63 genome editing. B6ES2 mESCs were injected into the BDF1-EGFP embryos at the 4–8 cell stage on F0 generation of the p63 genome editing. The chimeric embryos were cultured in KSOM/AA medium (Millipore, Billerica, MA, Catalog No. MR-101-D) until blastocyst stage and transferred into the uterus of pseudo-pregnant ICR mice. The sex of embryos was not considered. All donor mESCs are male.

### Genotyping

To extract the genomic DNA, the tails of p63 knockout mice were lysed with lysis buffer (Tris-HCl pH8.0 20 mM, NaCl 100 mM, five mM EDTA, 0.1% SDS) containing 400 μg/ml proteinase K (Sigma-Aldrich, St. Louis, MO, Catalog No. P6556) at 60 °C for 5 min and 98 °C for 2 min. To detect p63 mutation, the targeted region was amplified by PCR (Genotyping Forward primer: CACGTTTGTACAAGCCAGAACTTA, Genotyping Reverse primer: TCTTTTGGTCTTCCCGAGCCT) using

KOD-multi & Epi- (Toyobo, Osaka, Japan, Catalog No. KME-101). The amplicons were column-purified and directly sequenced by Sanger sequencing, or following the cloning on pUC19. The data were analyzed using TIDE[33]. When we found unexplainable results by TIDE analysis, we additionally verified the genotype using NGS-based genotyping. Illumina libraries were constructed with in-house custom index primers (Supplementary Data 2). The multiplex libraries were sequenced by Illumina HiSeq X (150 bp, paired end). The data were analyzed using Cas-analyzer[34] (Supplementary Fig. 2).

For genotyping of p63 knockout chimera, liver and spleen were hemolyzed in Ammonium-Chloride-Potassium (ACK) buffer and stained with CD45 antibody (1:100, clone 30-F11, eBioscience, Catalog No. 17-0451-82) for 30 min. Genotype was confirmed by host embryos-derived fibroblast (E9.5), fetal liver CD45+ cells (E14.5), and splenocytes (E18.5) following cell sorting by FACS aria II or III (BD).

### Flow cytometry

The skin was harvested from p63 knockout chimera and WT-chimera and digested in CnT-07 medium supplemented with 25 U/ml dispase (Gibco, Life Technologies, Carlsbad, CA, Catalog No. 17105-041) at 4 °C, 12–24 h. Epidermis isolated from the skin was digested with accutase (Innovative Cell Technologies, California, USA, Catalog No. AT-104) to prepare single-cell suspensions. Samples were stained with anti-CD45 (1:100, clone 30-F11, Biolegend, Catalog No. 103126), anti-CD117 (1:100, clone 2B8, eBioscience, Catalog No. 105814), and anti-CD49f (1:100, clone GoH3, eBioscience, Catalog No. 17-0495-82) antibodies for 30 min on ice. Spleens were dissociated into single cells by pipetting, hemolyzed in ACK buffer, and stained with anti-CD45 antibody (1:100, clone 30-F11, eBioscience, Catalog No. 17-0451-82) for 30 min.

Each sample was stained with Propidium iodide (PI) (Sigma-Aldrich, St. Louis, MO, Catalog No. P4170) or 4',6-diamidino-2-phenylindole (DAPI) (Invitrogen, Carlsbad, CA, Catalog No. D1306) and analyzed by CytoFLEX S (B75442, BECKMAN COULTER, California, USA). The chimerism of each tissue was calculated by referring to the EGFP signal using FlowJo software (BD biosciences, New Jersey, USA).

### Macroscopic analysis/Mosaic distribution calculation

The fetuses (E9.5, E14.5, E18.5) obtained by Cesarean section were photographed under a fluorescence microscope (LEICA M165 FC). The percentage of skin areas in WT-chimera or p63 knockout-chimeras at E18.5 were estimated according to the EGFP signal. The number of pixels of adjacent basal layers was measured in immunofluorescence staining images of WT chimera or p63 knockout chimera at E14.5. Total EGFP$^+$ or EGFP$^-$ areas were summed, respectively, and the frequency of occurrence per pixel (expected value) was calculated from Formula 1. Next, the probability of EGFP$^+$ and EGFP$^-$ regions in a sequence of adjacent regions were calculated from Formula 2. The mosaic distribution was quantitatively compared by taking the logarithm of the obtained expected values:

Formula 1:

$$\text{The probability that a pixel in the basal layer is EGFP}^+ \, (P_{\text{positive}})$$
$$= \text{total pixel value of EGFP}^+ \text{ area/total pixel value of all areas}$$
$$\text{Probability that a pixel in the basal layer is EGFP}^- \, (P_{\text{negative}})$$
$$= \text{total pixels in EGFP}^- \text{ area/total pixels in all areas}$$

Formula 2:

$$\text{Probability of existence of each region of the basal layer}$$
$$= \left(P_{\text{positive}} \text{ or } P_{\text{negative}}\right)^{\wedge} \text{(number of pixels in each region)}$$

### Histological analysis

Tissues were fixed in 4% paraformaldehyde at 4 °C for 24 h and then replaced with PBS. Fixed tissues were embedded in paraffin

blocks by a routine procedure, and HE-stained and unstained slides were prepared.

Tissue sections were deparaffinized in xylene and ethanol and autoclaved at 120 °C for 20 min in citric acid buffer at pH 6.0 for antigen retrieval.

Blocking was performed at room temperature for 30 min by MAXblock (Active motif, California, USA, Catalog No. 15252). Tissue sections were incubated with the primary antibody at 4 °C for 1–2 h or overnight, washed in PBS, and incubated with the secondary antibody at room temperature for 2 h. Primary antibodies: mouse anti-p63 (1:100, clone D-9, sc-25268, Santa Cruz), rabbit anti-Cytokeratin 1 (1:200, polyclonal, ab93652, Abcam), rabbit anti-Cytokeratin 5 (1:200, clone EP1601Y, ab52635, Abcam), rabbit anti-Cytokeratin 10 (1:200, clone EP1607IHCY, ab76318, Abcam), rabbit anti-Cytokeratin 14 (1:200, clone EPR17350, ab181595, Abcam), rabbit anti-Cytokeratin 8/18 (1:200, clone EP1628Y, ab53280, Abcam), rabbit anti-Loricrin (1:200, polyclonal, ab85679, Abcam), rabbit anti-Involucrin (1:200, clone EPR13054, ab181980, Abcam) and goat anti-GFP (1:200, polyclonal, ab6673, Abcam). Secondary antibodies: Donkey anti-goat IgG Alexa Fluor 488 (1:1000, A11055: Invitrogen), Donkey anti-mouse IgG Alexa Fluor 568 (1:1000, A10037: Invitrogen), Donkey anti-rabbit IgG Alexa Fluor 568 (1:1000, A10042: Invitrogen) and Donkey anti-rabbit IgG Alexa Fluor 647 (1:1000, A31573: Invitrogen). Samples were washed with PBS and mounted with Dako fluorescent mounting medium (DAKO, CA, United States, Catalog No. S3023). TUNEL staining (Click-iT™ TUNEL assay, C10619, Invitrogen) was performed according to the manufacturer's protocol. A sample without TUNEL staining was prepared as a negative control, and a sample treated with DNase was prepared as a positive control. As for the human skin segment, frozen sections were prepared by immersing the tissue in OCT compound (Sakura Finetek Japan, Tokyo, Japan, Catalog No. 4583), cooling at −80 °C, slicing to 7 μm using a cryostat (Leica, CM3050S), and performing antibody staining. Images were acquired using confocal laser scanning microscopy (FV3000; Olympus, Tokyo, Japan).

### Single cell RNAseq
The skin of WT (C57BL/6N background) fetuses at E14.5 was soaked in dispase and separated into the epidermis and dermis. The single cells of the WT epidermis by treated with trypsin. The p63 knockout fetuses (C57BL/6N background) were directly digested with trypsin. Samples were stained with following antibodies: anti-mouse CD45 (1:100, clone 30-F11, eBioscience, Catalog No. 103126), TER119 (1:100, clone TER-119, eBioscience, Catalog No. 25-5921-82), anti-CD140a (1:100, clone APA5, Biolegend, Catalog No. 135908). The single cell of the PI⁻/CD45⁻/TER119⁻/CD140a⁻ population was sorted by FACS aria II or III (BD). The cDNA library was prepared by SMART-Seq HT (Takara Bio Inc., Shiga, Japan, Catalog No. 634438) and purified with AMPure XP beads (BECKMAN COULTER, California, USA, Catalog No. A63881). We screened samples by qPCR with Taqman Fast advanced Master Mix using Quant studio 7 (Applied Biosystems, Massachusetts, USA, Catalog No. 4444557). *Gapdh* (Forward primer: TGGAGAAACCTGCCAAGTATG, Reverse primer: TGGGAGTTGCTGTTGAAGTC, Probe: CATCAAGAAGGTGGTGAAGCAGGC) was used as a reference gene for sample screening, and samples with significant amplification of *Krt5* (Forward primer: TGAACCGAATGATCCAGAGG, Reverse primer: GCTCTGTCAGCTTGTTTCTG, Probe: AACGTCAAGAAGCAGTGTGCCAAC) or *Krt8* (Forward primer: GATGCAGAACATGAGCATTCA, Reverse primer: CATTCCGTAGCTGAAGCCAG, Probe: CGGCTACTCAGGAGGACTGAGTTCA) were selected (*Krt5* for WT keratinocytes, *Krt8* for p63 knockout keratinocytes). We also confirmed that *Pdgfra* (Forward primer: AAAAGCAGGCTCTCATGTCT, Reverse primer: AGTAGTTGACCAAATCCCCA, Probe: TGCACCAAGTCAGGTCCCATTTACA) was not significantly amplified. Using selected samples, we prepared to Illumina library by Nextera XT (Illumina, California, USA, Catalog No. FC-131-1024) with a slightly-modified protocol and sequenced by Hiseq (150 bp paired end). At each step, we performed a quality check of the cDNA and Illumina libraries by Tapestation (Agilent, California, USA). We first mapped the data to mm10 by HISAT2[35] (Galaxy Version2.1.0+galaxy5) on Galaxy[36] (version20.05) and counted by Feature Counts[37] (Galaxy Version 1.6.4+galaxy2) based on the GTF file of Ensemble[38], version-release.97. Transcript integrity number (TIN) and Gene Body Coverage were analyzed by RSeQC[39] (Galaxy Version 2.6.4.1 and Galaxy Version 2.6.4.3) to exclude low-quality samples. The count was normalized by DESeq2[40] (version1.38.2) and analyzed for differentially expressed genes (DEGs). The $p$ value was calculated using a two-sided Wald test. We visualized the data using iDEP[41] (version0.96) or R[42] (version3.6.0 and 4.2.2). Boxplots for visualization of gene expression are generated using the ggplotgui package (https://site.shinyserver.dck.gmw.rug.nl/ggplotgui/).

### Skin grafting
Skin grafts were harvested from E18.5 fetuses (see Fig. 3d). Adipose tissue was removed from the harvested skin. The back skin of the recipient mice was excised, and the skin grafts were sutured with 5-0 nylon. The skin grafts were fixed by tie-over fixation with sutures and gauze, and then adhesive tape was applied around the trunk. Seven days later, the tie-over fixation was removed, and photographs were taken every 2–3 days to confirm the viability of the skin grafts. Furthermore, we performed the histological analysis with biopsied samples of engrafted skin grafts at 1.5 months and 3 months after transplantation.

### In utero injection
Pregnant mice at E13.5 implanted with p63 knockout fertilized eggs were abdominally opened under inhalation anesthesia.

The cell suspension of HaCaT-EGFP ($1.0 \times 10^7$ cells/100 μl CnT-07) was injected transuterine into the amniotic cavity with a 27G needle syringe. We obtained the fetuses at E18.5 by Cesarean section. We performed macroscopic fluorescence analysis and histological analysis.

### Statistics and reproducibility
All experiments are performed at least three independent trials. Statistical analysis was performed with GraphPad Prism v.8.4.3. The comparison between multiple groups was performed by Repeated Measures ANOVA (Friedman's test) and post hoc test by Dunn's multiple comparison test. A $p$ value less than 0.05 was considered significant.

### Reporting summary
Further information on research design is available in the Nature Portfolio Reporting Summary linked to this article.

## Data availability
All relevant data supporting the key findings of this study are available within the article and its Supplementary Information files or from the corresponding author upon request. Mouse genome mm10 is used as reference. The scRNA-Seq data used in this study are available in SRA database under BioProject accession number PRJNA1056324. The processed scRNA-Seq data are available in the Gene Expression Omnibus database under accession number GSE252023. Source data are provided with this paper.

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

## Acknowledgements

We thank all members of the Nakauchi lab for discussion. We are also particularly grateful to Ms. Hiroko Tsukui for technical support, Ms. Kyoko Okada and Ms. Asami Miyamoto for secretarial support. We appreciate Ms. Yumiko Ishii and Ms. Azusa Fujita (FACS Core Laboratory, Institute of Medical Science, University of Tokyo) for performing flow cytometry and Ms. Tomoko Ando (Pathology Core Laboratory, Institute of Medical Science, University of Tokyo) for performing pathological specimen preparation and staining. We are grateful to Dr. Toshihiro Kobayashi (Institute of Medical Science, University of Tokyo), Dr. Ryuichi Azuma (National Defense Medical College), Dr. Tomoharu Kiyosawa (National Defense Medical College) for helpful discussions and suggestions. The authors used Biorender (http://biorender.com) in the creation of the figures. This work was supported by grants from the Centers for Clinical Application Research on Specific Disease/Organ (to H.Nakauchi) of the Research Center Network for Realization of Regenerative Medicine, funded by the Japan Agency for Medical Research and Development (AMED) under Grant Number JP22bm1004002; Grant-in-Aid for Scientific Research (B) (to E.M.), funded by the Japan Society for the Promotion of Science (JSPS) KAKENHI under Grant Number JP20H03638; Grant-in-Aid for Scientific Research (C) (to N.M.), funded by JSPS KAKENHI under Grant Number 21K07837; A.Y. is supported by the University of Tokyo Excellent Young Researcher system.

## Author contributions

Conceptualization: H.Nagano, N.M., and H.Nakauchi; Methodology: M.Kasai, H.S., and E.M.; Validation: A.Y., H.M., and M.W.; Data curation: H.Nagano, N.M., H.Y., M.Kano, F.S., M.Kasai., H.S., and E.M.; Writing—original draft: H.Nagano and N.M.; Writing—review & editing: F.S. and H.Nakauchi; Investigation: H.Nagano, N.M., H.Y., M.Kano, M.Kasai, H.S., and E.M.; Funding acquisition: N.M., E.M., and H.Nakauchi; Resources: M.Kasai, H.S., and E.M.; Project administration: H.Nagano, N.M., H.M., and H.Nakauchi; Supervision: N.M. and H.Nakauchi.

## Competing interests

H.Nakauchi is a co-founder and shareholder in ReproCELL, Megakaryon, Century Therapeutics, and Celaid Therapeutics. M.W. is a co-founder and shareholder in Megakaryon and Celaid Therapeutics. N.M., H.Nakauchi, H.Nagano, H.M., M.W., H.S. are inventors (The University of Tokyo) on a patent for SKIN TISSUE (WIPO Patent Application PCT/JP2022/41189). The remaining authors declare no competing interests.
