## [Peer Review File · Nature Communications]

REVIEWER COMMENTS

Reviewer #1 (Remarks to the Author):

Utilizing human stem cells to generate skin with functionality and transplantability is a pivotal goal to address the shortage of skin grafts. The current bottleneck in this domain is the inability to obtain artificial skin with skin adnexa such as hair follicles, sweat glands, and nails on a large scale. The p63 is crucial for the differentiation of keratinocytes and other stratified epithelia. In this study, the authors micro-injected embryonic stem cells (ESCs) from mice, tagged with green fluorescent protein (EGFP), into early-implanted embryos at the 4-8 cell stage with p63 knockout. Out of 79 transplanted embryos, 8 chimeric newborns with p63 gene knockout were obtained. Upon digesting the embryonic skin to single cells, flow analysis confirmed that cells expressing the mature keratinocyte marker CD49f were almost entirely derived from donor cells. At the E18.5 stage, the larger the skin area obtained, the higher the global chimerism rate of donor cells. The donor PSC-derived keratinocytes in the p63 gene knockout chimeras formed well-differentiated and stratified epidermis similar to the natural keratinocytes of wild-type (WT) mice. Hair follicles also originated from donor PSCs. Statistical results revealed that PSC-derived keratinocytes formed more continuous segments, indicating that PSC-derived keratinocytes competitively took over the epidermal niche on the dermis by attenuating the p63 knockout SERs, thereby forming segmented clusters without intercalated SER interruptions. Single-cell RNA-Seq of SER and WT keratinocytes showed that p63 gene knockout host SER might be pushed out of the niche by PSC source keratinocytes in the chimera due to lower cell adhesiveness. Next, the researchers explored the engraftment potential of skin grafts derived from p63 knockout chimeras. Semi-autologous skin grafting was performed by transplanting the grafts onto recipient mice of varied genetic backgrounds. Remarkably, the semi-autologous skin grafts from p63 knockout chimeras exhibited long-term engraftment, akin to autologous grafts. This suggested that a match in the epidermal major histocompatibility complex (MHC) was sufficient for successful engraftment, whereas dermal cell MHC matching was not imperative. In the concluding experiments, the researchers aimed to create semi-humanized skin by injecting human keratinocytes into the amniotic cavity of p63 knockout mice. Formation of human skin segments was observed, corroborated by macroscopic and fluorescence imaging. Immunofluorescence staining unveiled the presence of human keratinocytes expressing p63 and Krt14 in the developed skin segments. In summary, this study underscores the significance of p63 in skin development and skin graft formation. The findings hint at abnormalities in the basal cell layer and hindered differentiation of keratinocytes, resulting from p63 knockout. Skin grafts from p63 knockout chimeras can engraft successfully, highlighting the crucial role of epidermal MHC matching for graft survival. Moreover, the study unveils the potential for engendering semi-humanized skin by injecting human keratinocytes into p63 knockout mice. These insights augment our comprehension of skin development and bear implications for regenerative medicine and tissue engineering.

Below are several specific comments:

1. You have designed a gRNA targeting Exon 5 of p63, and typically, the editing efficiency ranges between 50%-60%. In Supplementary Fig. 1a, the efficiency of KO phenotype(+) reached 77.3%. Is this because heterozygous p63 also exhibits a KO phenotype? Additionally, how do you determine the KO phenotype?

What percentage of skin damage is required to define a KO phenotype? It may be beneficial to use 3-4 gRNAs targeting p63 to achieve closer to 100% knockout efficiency (PMID: 28585534).

2. Does the knockout of p63 affect limb development? In the compensatory embryonic limb tissues, what is the proportion of donor cells?

3. How is Global chimerism quantified in Figure 1f? In the 8 p63 gene knockout chimeric newborns obtained, there were EGFP cells chimerized into p63 unknocked embryos. How did you distinguish between them? For instance, did you sort out non-green cells for genotyping?

4. Cell competition usually induces apoptosis in surrounding cells. Since the number of TUNEL-positive cells in p63 knockout embryos and p63 knockout chimeras was not significantly different from the number of TUNEL-positive cells in WT embryos, can it be concluded that this is a result of cell competition?

Reviewer #2 (Remarks to the Author):

This study showed that donor pluripotent stem cells derived keratinocytes are localized in the epidermis eliminating p63 knockout keratinocytes. In addition, semi-humanized skin segments were observed after human keratinocytes injection into the amniotic cavity of p63 knockout mice embryos.

The technique of niche encroachment allow to generate human skin graft by injecting human keratinocytes in amniotic cavity at E18.5 using livestock animals to treat skin defects. However, only small segment of sheet expressing keratin 14 are observed. The human epidermis does not seem to stratified properly. Only injection of mouse pluripotent stem cells into 4-8 cells stage allow the formation of pluristratified epidermis with EGFP PSC derived keratinocytes but interspersed by host keratinocytes.

Human:animal chimeric skin graft remains an issue at ethical level.

Additional experiments are require to demonstrate the possibility to generate functional stratify human epidermis such immunostaining for keratin 10, involucrin keratin 14 and keratin 5 after injection of human keratinocytes in amniotic cavity.

This work is original and showed for the first time that competition between donor cells and p63 knockout keratinocytes. To identify how PSC-derived cells invade the host embryos' epidermal niche by eliminating host keratinocytes during embryogenesis authors perform a single cell RNA seq. They identify a down regulation of gene expression of protein implicated in the hemidesmosomes. This results need to be completed at protein levels by immunofluorescence analyses.

The work need to be improved and completed to be used to the field of skin cell therapy to ovoid chimeric skin grafting in human.

Reviewer #3 (Remarks to the Author):

##What are the noteworthy results?

Nagano et al generated a model of skin grafts containing appendages and dermis. Models of skin containing appendages such as sebaceous glands and dermis are very much needed for skin modelling and regenerative medicine. This work describes some steps towards this direction.

##Will the work be of significance to the field and related fields? How does it compare to the established literature? If the work is not original, please provide relevant references.

The authors explore an interesting chimeric model to complement a mouse line incompetent for epidermis with PSC. They show important complementation for generating epidermis and its appendages.

The authors measure immunogenicity of the dermis vs epidermis which indicates interesting conclusions for future bioengineering.

The authors perform an intriguing human complementation using immortalized keratinocytes which suggest interesting future experiments for semi-humanized skin.

##Does the work support the conclusions and claims, or is additional evidence needed?

The authors make the very interesting observation regarding compatibility dependence on epidermis components. I was wondering whether how much of this effect is due to dermis cells competition of recipient to donor. I believe that a quantification of donor vs recipient cells would help clarifying.

##Are there any flaws in the data analysis, interpretation and conclusions? Do these prohibit publication or require revision?

Could the authors elaborate a bit more on the prospects of semi-humanized skin in larger animals?
Would this human cells require genetic modification to increase developmental compatibility?

The authors discuss the reduced ethical concerns regarding chimera generation with method. I agree that the probability to neural contribuion or germinal contribution is low, perhaps still further steps may be required such as human cells with selective KO to completely prevent neuronal/germinal contribution.

##Is the methodology sound? Does the work meet the expected standards in your field?

In general yes. A minor comment to Supp Figure 1 would be that genotyping based on TIDE methodology may not be optimal for accurate genotyping. NextGen based methods such as crispresso, crisprseek, crispr-a would provide a more sensitive measurement.

##Is there enough detail provided in the methods for the work to be reproduced?

yes

We would like to thank the reviewers for the thorough review of our manuscript and the constructive comments. We have replied to each reviewer's comment. The reviewer's comments are formatted in bold and italics. The revised portions of the manuscript are highlighted in yellow.

Reviewer #1 (Remarks to the Author):

1. You have designed a gRNA targeting Exon 5 of p63, and typically, the editing efficiency ranges between 50%-60%. In Supplementary Fig. 1a, the efficiency of KO phenotype(+) reached 77.3%. Is this because heterozygous p63 also exhibits a KO phenotype? Additionally, how do you determine the KO phenotype? What percentage of skin damage is required to define a KO phenotype? It may be beneficial to use 3-4 gRNAs targeting p63 to achieve closer to 100% knockout efficiency (PMID: 28585534).

We appreciate the reviewer's suggestion. As shown in Supplementary Fig. 1a, 17 of 22 fetuses had the knockout (KO) phenotype (KO phenotype (+)). Although, all 22 live fetuses had some mutant alleles (Supplementary Fig. 1b), we only enrolled fetuses with both the KO phenotype and complete loss-of-function genotypes. Eleven of 17 fetuses with the KO phenotype had bi-allelic loss-of-function mutations. The bi-allelic loss-of-function mutations were defined

as the genotypes composed of either of out-of-frame mutations or large deletions. The remaining 6 of 17 fetuses with the KO phenotype contained in-frame mutations.

These in-frame mutations could still have caused the KO phenotype because the gRNAs targeted a critical core domain in the GOI (Kano et al., PNAS. 2023). This explains the higher number of fetuses with a KO phenotype than those with a bi-allelic loss-of-function genotype.

The p63 KO phenotype of the fetus was defined according to transparent skin on the whole body and shortened limbs. Therefore, in this experiment, only fetuses with 100% skin damage on the whole body were considered to have the KO phenotype. Regarding heterozygosity, only one of the 22 fetuses had a heterozygous p63 KO genotype (Supplementary Fig. 1b, #7) for which the phenotype was normal as expected (KO phenotype (-)).

We added detailed information on genetically modified fetuses in Supplementary Fig 1b, and revised the original manuscript as follows.

Page 6-7, Line #141-151:

“Seventeen of 22 live fetuses showed knockout phenotype, and 11 of 17 phenotypically Knockout fetuses had bi-allelic loss-of-function genotypes (Supplementary Fig. 1a). Although multiple gRNAs could achieve higher editing efficiency (Zuo et al., Cell Res. 2017), we introduced single gRNAs because small InDels

are precisely quantified by simple genotyping methods (Supplementary Fig. 2a-d).

Some fetus showed knockout phenotypes even though their genotype contained small in-frame mutations, probably because the gRNA was designed to target a core domain of p63 (Supplementary Fig. 1b) (Kano et al., PNAS. 2023). However, we excluded these fetuses from the following experiments because their genotype is not guaranteed to produce a knockout phenotype.”

2. Does the knockout of p63 affect limb development? In the compensatory embryonic limb tissues, what is the proportion of donor cells?

Thank you for pointing this out. As the reviewer mentioned, the p63 knockout fetus showed shortening of the limbs, while the p63 knockout chimera had restored limb-size. We performed a histological analysis of the limb of the p63 knockout chimera. In summary, the restored forearm was not composed completely of donor PSC-derived cells. Although the epidermis was composed of 100% donor PSCs-derived cells, the deeper tissues such as the dermis, subcutaneous tissue, muscles, and bones were in a chimeric state with donor PSC-derived cells and host embryo-derived cells (Supplementary fig. 3a-i).

We have shown the limb histological analysis in a new supplementary figure, supplementary

Fig. 3. We also revised the original manuscript as follows.

Page 8, Line #173-176:

“Some fetuses showed normal limbs and craniofacial formation. In the restored forearms, the epidermis was comprised of donor PSC-derived cells, while subcutaneous and bone tissues were chimeric (Supplementary Fig. 3a-i).”

3. How is Global chimerism quantified in Figure 1f? In the 8 p63 gene knockout chimeric newborns obtained, there were EGFP cells chimerized into p63 unknocked embryos. How did you distinguish between them? For instance, did you sort out non-green cells for genotyping?

We apologize for the confusing description. Quite simply, we defined blood chimerism as global chimerism. To estimate global chimerism, white blood cells are the best tissue for the analysis. We can collect them from peripheral blood, spleens, or fetal livers with minimal sampling bias compared to other cell lineages of solid organs. It correlates well with organs' chimerism in allogeneic chimeras (Yamaguchi & Sato et al. Sci Rep. 2018).

We sorted host embryo-derived cells by FACS for genotyping the p63 knockout chimeras.

The sorting of host embryo-derived cells and donor PSC-derived cells is based on the absence

or presence of EGFP, respectively.

We added the citation above to the result section and revised the method description as follows.

Page 8, Line #170-171:

“The total skin area of p63 knockout chimeras at E18.5 increased logarithmically with global chimerism assumed by the splenocyte chimerism (Yamaguchi & Sato et al. Sci Rep. 2018).”

Page 23, Line #535-537:

“Genotype was confirmed by host embryos-derived fibroblast (E9.5), fetal liver CD45+ cells (E14.5), and splenocytes (E18.5) following cell sorting by FACS aria II or III (BD).”

4. Cell competition usually induces apoptosis in surrounding cells. Since the number of TUNEL-positive cells in p63 knockout embryos and p63 knockout chimeras was not significantly different from the number of TUNEL-positive cells in WT embryos, can it be concluded that this is a result of cell competition?

As the reviewer pointed out, the canonical “cell competition” directly induces apoptosis of

loser cells adjacent to winner cells. On the other hand, apoptosis-independent cell competition called apical extrusion is known to occur in the epithelium of luminal organs such as the intestine. In apical extrusion, the loss of cell-cell adhesion causes loser cells to migrate to the outermost layers and become physically removed from the body, leading to cell death called anoikis. Recently, a new mechanism of cell competition due to a loss of cell-ECM adhesion has been discovered by Nishimura in the skin (Liu et al. Nature. 2019). Nishimura's cell competition removes the rare senescent cells in the skin. We found that the loss of cell-ECM adhesion mediates cell competition during embryogenesis, resulting in massive tissue replacement in p63 knockout chimera.

We revised the original manuscript as follows.

Page 17, Line #396-400:

“The canonical cell competition directly induced apoptosis of loser cells adjacent to winner cells, but apoptosis independent-cell competition was also reported. Apical extrusion in the luminal organs removes the outlier cells from the contiguous epithelium. However, it is mediated by disruption of cell-cell adhesion.”

Reviewer #2 (Remarks to the Author):

This study showed that donor pluripotent stem cells derived keratinocytes are localized in the epidermis eliminating p63 knockout keratinocytes. In addition, semi-humanized skin

segments were observed after human keratinocytes injection into the amnionic cavity of p63 knockout mice embryos.

The technique of niche encroachment allow to generate human skin graft by injecting human keratinocytes in amionic cavity at E18.5 using livestock animals to treat skin defects.

However, only small segment of sheet expressing keratin 14 are observed. The human epidermis does not seem to stratified properly. Only injection of mouse pluripotent stem cells into 4-8 cells stage allow the formation of pluristratied epidermis with EGFP PSC derived keratinocytes but interspersed by host keratinocytes.

Human:animal chimeric skin graft remains an issue at ethical level.

Additional experiments are require to demonstrate the possibility to generate fonctional stratify human epidermis such immunostaining for keratin 10, involucrin keratin 14 and keratin 5 after injection of human keratinocytes in amniotic cavity.

Thank you for your constructive suggestion. We have performed additional experiments to observe whether in utero injected HaCaTs differentiate on p63 knockout dermis. Our initial expectation was that the gestation period of the mice was too short for the injected HaCaTs to attach and fully stratify while differentiating. Because it was only five days from the time of injection (E13.5) to the time of sample collection (E18.5). Based on a previous report

(Schoop et al. *J Invest Dermatol.* 1999) during which HaCaTs sheeted on collagen gels were transplanted into the back of a mouse, the expression of differentiation markers such as KRT1/10 only begins to be observed on day 4, and it takes 1-2 weeks to observe the expression of these markers clearly. Late differentiation markers such as Loricrin take even longer, reportedly. Since our experiments used single-cell dissociated human keratinocyte cell lines, we predicted that the expression of differentiation markers would take even longer than using HaCaTs sheets. However, surprisingly, the injected HaCaTs were positive for differentiation markers such as KRT1/10 and late differentiation markers such as IVL and LORICRIN (Fig. 4c-h). Of course, they do not show complete layer formation or differentiation marker patterns, but we believe that the extension of incubation period in future large-animal experiments will solve this problem. In any case, the amniotic cavity human keratinocyte injection experiment in Figure 4 demonstrated that the main mechanism of niche encroachment, epidermal replacement by cell competition, works effectively between human keratinocytes and xenogeneic p63 knockout embryos. This experiment was performed to demonstrate the proof-of-concept of the mechanism. Experiments to generate practical semi-humanized skin in livestock with longer gestation periods are in the planning stages and beyond the purpose of this study.

We replaced Fig. 4c to the 4c-h and revised the original manuscript as follows.

Page 13-14, Line #311-313:

“Differentiation markers such as KRT1/10, LORICRIN, and IVL were positive in the outer layer, including the suprabasal layer (Fig. 4d-i), suggesting that HaCaT was stratified and differentiated on the p63 knockout dermis.”

This work is original and showed for the first time that competition between donor cells and p63 knockout keratinocytes. To identify how PSC-derived cells invade the host embryos' epidermal niche by eliminating host keratinocytes during embryogenesis authors perform a single cell RNA seq. They identify a down regulation of gene expression of protein implicated in the hemidesmosomes. This results need to be completed at protein levels by immunofluorescence analyses. The work need to be improved and completed to be used to the field of skin cell therapy to ovoid chimeric skin grafting in human.

We thank the reviewer for the advice. Following the single cell RNAseq results, we attempted immunofluorescence analysis for hemidesmosome-associated proteins using antibodies against COL4 (Abcam, ab19808), COL17A1 (Abcam, ab186415), LAMININ (Abcam, ab11575),

etc. Unfortunately, none of them worked efficiently in mice skin. It is known that anti-mouse antibodies are limited in commercial availability. Sekiguchi et al. have successfully stained with their anti-mouse antibodies (Manabe et al. PNAS. 2008.), but their antibodies were not available at this time. As for the anti-human antibodies, several antibodies are commercially available, and we will be able to prove this when we do it with large animals in the future.

Reviewer #3 (Remarks to the Author):

##What are the noteworthy results?

Nagano et al generated a model of skin grafts containing appendages and dermis. Models of skin containing appendages such as sebaceous glands and dermis are very much needed for skin modelling and regenerative medicine. This work describes some steps towards this direction.

##Will the work be of significance to the field and related fields? How does it compare to the established literature? If the work is not original, please provide relevant references.

The authors explore an interesting chimeric model to complement a mouse line incompetent for epidermis with PSC. They show important complementation for generating epidermis and its appendages.

The authors measure immunogenicity of the dermis vs epidermis which indicates interesting conclusions for future bioengineering.

The authors perform an intriguing human complementation using immortalized keratinocytes which suggest interesting future experiments for semi-humanized skin.

##Does the work support the conclusions and claims, or is additional evidence needed?

The authors make the very interesting observation regarding compatibility dependence on epidermis components. I was wondering whether how much of this effect is due to dermis cells competition of recipient to donor. I believe that a quantification of donor vs recipient cells would help clarifying.

We agree with the reviewer that the relationship between dermal chimerism and the graft survival rate in the skin grafting experiments is important. Since it is difficult to control the chimerism, we performed a subgroup analysis of 8 animals with fibroblast chimerism ranging from 20-40% (WT chimera vs. p63 knockout chimera). The results showed that the average survival period for the WT chimera was 11.5 days, while the average survival period for the p63 knockout chimera was 91 days (grafts survived until the end of the observation period). This result supports the conclusion that the epidermis has a greater effect on grafts than the dermis.

We appended supplementary Fig. 8 and revised the original manuscript as follows.

Page 13, Line #291-295:

“To exclude the possibility of a high graft survival rate achieved by high dermal cell chimerism, we selected fetuses with dermal chimerism in the 20-40% range in WT and p63 knockout chimera. The subgroup analysis showed that the graft survival period was predominantly longer in the p63 knockout chimera than WT chimera (Supplementary Fig. 8a-c).”

##Are there any flaws in the data analysis, interpretation and conclusions? Do these prohibit publication or require revision?

Could the authors elaborate a bit more on the prospects of semi-humanized skin in larger animals? Would this human cells require genetic modification to increase developmental compatibility?

We agree with the reviewer that genetic modification of human cells may be advantageous for generating transplantable organs. We have also confirmed that minor immunological reactions occur after birth in xenogeneic chimeras (Yamaguchi et al. Nature. 2017). Therefore, we confirmed that modulation of genes involved in immunity (*IL2RG*, *Rag2*) in both donor cells and host embryos could suppress immunological reactions and allow more efficient organ generation (PCT/JP2018/002178, WO2018/13/13).

We revised the original manuscript as follows.

Page 18, Line #423-431:

“In this study, we successfully generated partially differentiated semi-humanized skin by niche encroachment. Human keratinocyte cell lines, which were injected into the mouse amniotic cavity, engrafted onto the embryos’ skin and expanded on the p63 knockout mouse dermis. The human cells formed a sheet-like and multi-layered structure with cytodifferentiation. Niche encroachment is promising even in the xenogeneic condition and during post-implantation organogenesis. Though postnatal immunological responses (Yamaguchi et al. Nature. 2017) may also be a problem in xenogeneic chimeras, modulation of genes involved in immunity (*IL2RG*, *RAG2*, etc.) may support efficient organ generation in vivo.”

The authors discuss the reduced ethical concerns regarding chimera generation with method. I agree that the probability to neural contribution or germinal contribution is low, perhaps still further steps may be required such as human cells with selective KO to completely prevent neuronal/germinal contribution.

We agree that we may need to perform additional gene knock out of human donor stem cells to completely prevent contribution in neuronal/germinal tissue. This has already been investigated and validated by another researcher in our lab (in submission).

We revised the original manuscript as follows.

Page 19, Line #445-447:

“Gene knockouts related to neuronal (*OTX2*) or germinal (*PRDM14*(Kobayashi et al. Nat Commun. 2021)) cell differentiation may further minimize the risk of human stem cell contribution in the brain or gonad.”

##Is the methodology sound? Does the work meet the expected standards in your field?

In general yes.

A minor comment to Supp Figure 1 would be that genotyping based on TIDE methodology may not be optimal for accurate genotyping. NextGen based methods such as crispresso, crisprseek, crispr-a would provide a more sensitive measurement.

We agree with the reviewer that NGS-based genotyping is an accurate and powerful method and more preferable in large InDels. In this experiment, we used only one gRNA to edit the genome of fertilized eggs, so the number of mutant alleles is limited from one to four, and

most of them are small InDels. Therefore, TIDE analysis was sufficient for most of the embryos. When we found unexplainable results by TIDE analysis following sanger sequencing, we performed NGS-based genotyping and/or the classical cloning into plasmid to identify large deletions.

We added NGS analysis in a new supplementary figure, supplementary Fig. 2. We also revised the original manuscript as follows.

Page 6-7, Line #143-146:

“Although multiple gRNAs could achieve higher editing efficiency¹², we introduced single gRNAs because small InDels are precisely quantified by simple genotyping methods (Supplementary Fig. 2a-d).”

Page 22-23, Line #528-532:

“When we found unexplainable results by TIDE analysis, we additionally verified the genotype using NGS-based genotyping. Illumina libraries were constructed with in-house custom index primers (supplementary table 3). The multiplex libraries were sequenced by Illumina HiSeq X (150 bp, paired end). The data were analyzed using Cas-analyzer³³ (Supplementary Fig. 2).”

REVIEWERS' COMMENTS

Reviewer #1 (Remarks to the Author):

After carefully reviewing the revisions made by the authors in response to the concerns I raised in the previous round of review, I am pleased to find that all the issues have been adequately and satisfactorily addressed. Moreover, I have noted a significant overall improvement in the quality of the manuscript. Therefore, based on the current quality of the manuscript, I believe the article is now suitable for publication in this journal. I am confident that this paper will make a valuable contribution to academic discourse in its field.

Reviewer #3 (Remarks to the Author):

The authors addressed well all the previous points raised. No more comments on my side.